# Methylomic analysis identifies *C11orf87* as a novel epigenetic biomarker for GI cancers

**Mita T. M. T. Tran**[1,2,3☺]**, Kun-Tu Yeh**[4,5☺]**, Yu-Ming Chuang**[1,2,3☺]**, Po-Yen Hsu**[1,2,3]**, Jie-Ting Low**[1,2,3,6]**, Himani Kumari**[1,2,3]**, Yu-Ting Lee**[1,7]**, Yin-Chen Chen**[6]**, Wan-Hong Huang**[1,2,3]**, Hongchuan Jin**[8]**, Shu-Hui Lin**[4,9]***, Michael W. Y. Chan**[1,2,3]***

**1** Department of Biomedical Sciences, National Chung Cheng University, Min-Hsiung, Chia-Yi, Taiwan, **2** Epigenomics and Human Disease Research Center, National Chung Cheng University, Min-Hsiung, Chia-Yi, Taiwan, **3** Center for Innovative Research on Aging Society (CIRAS), National Chung Cheng University, Min-Hsiung, Chia-Yi, Taiwan, **4** Department of Surgical Pathology, Changhua Christian Hospital, Changhua, Taiwan, **5** School of Medicine, Chung Shan Medical University, Taichung, Taiwan, **6** Division of Gastroenterology, Chang Gung Memorial Hospital, Chia-Yi, Taiwan, **7** Department of Hematology and Oncology, Ditmanson Medical Foundation Chiayi Christian Hospital, Chiayi, Taiwan, **8** Laboratory of Cancer Biology, Key Lab of Biotherapy in Zhejiang, Sir Run Run Shaw Hospital, Medical School of Zhejiang University, Hangzhou, China, **9** Department of Medical Laboratory Science and Biotechnology, Central Taiwan University of Science and Technology, Taichung, Taiwan

☺ These authors contributed equally to this work.
* 74630@cch.org.tw (SHL); biowyc@ccu.edu.tw (MWYC)

**Data Availability Statement:** The methylation array data has been deposited into in the Gene Expression Omnibus database (accession number: GSE109541).

## Abstract

Gastric cancer is one of the leading causes of cancer death worldwide. Previous studies demonstrated that activation of STAT3 is crucial for the development and progression of gastric cancer. However, the role of STAT3 in neuronal related gene methylation in gastric cancer has never been explored. In this study, by using DNA methylation microarray, we identified a potential STAT3 target, *C11orf87*, showing promoter hypomethylation in gastric cancer patients with lower STAT3 activation and AGS gastric cancer cell lines depleted with STAT3 activation. Although *C11orf87* methylation is independent of its expression, ectopic expression of a constitutive activated STAT3 mutant upregulated its expression in gastric cancer cell line. Further bisulfite pyrosequencing demonstrated a progressive increase in DNA methylation of this target in patient tissues from gastritis, intestinal metaplasia, to gastric cancer. Intriguingly, patients with higher *C11orf87* methylation was associated with better survival. Furthermore, hypermethylation of *C11orf87* was also frequently observed in other GI cancers, as compared to their adjacent normal tissues. These results suggested that *C11orf87* methylation may serve as a biomarker for diagnosis and prognosis of GI cancers, including gastric cancer. We further postulated that constitutive activation of STAT3 might be able to epigenetically silence *C11orf87* as a possible negative feedback mechanism to protect the cells from the overactivation of STAT3. Targeted inhibition of STAT3 may not be appropriate in gastric cancer patients with promoter hypermethylation of *C11orf87*.

**Funding:** This study was supported by grants from the Ministry of Science and Technology, Taiwan (MOST 106-2314-B-194-001-MY3; 107-2314-B-194-001; 108-2314-B-194-001; 108-2314-B-194-003- MY2) to MWYC, Changhua Christian Hospital, Taiwan (106-CCH-IRP-064) to SHL, and the Center for Innovative Research on Aging Society (CIRAS) from The Featured Areas Research Center Program within the framework of the Higher Education Sprout Project by Ministry of Education (MOE) in Taiwan.

**Competing interests:** The authors have declared that no competing interests exist.

## Introduction

Gastric cancer is the third leading cause of cancer death worldwide [1]. Infection with *Helicobacter pylori* (*H. pylori*), a gram-negative bacteria, is an important risk factor for gastric cancer [2]. Particularly, cytotoxin-associated gene A (CagA) positive *H. pylori*, which induces inflammation and activation of JAK/STAT signaling, have a higher risk of developing gastric cancer [3, 4]. Previous studies also confirmed that STAT3 activation is decisive for the initiation, and progression in gastric cancer patients [5–7]. Several studies, including ours, demonstrated that STAT3 activation may confer aberrant epigenetic modifications in gastric epithelial cells [8–12]. However, the clinical significance of these modifications is not fully explored.

Abnormal DNA methylation is being considered as a hallmark of cancers because of a crucial mechanism in regulating transcription [13]. In such, aberrant promoter DNA methylation patterns in human cancers may be linked to the specific signaling pathways that are dysregulated in response to unique carcinogens exposed to specific tumor types [14]. Additionally, DNA methylation alteration has related to cancer development [15, 16].

We have previously demonstrated methylation of multiple STAT3 targets in gastric cancer patients [8–10]. In this study, we further identified a putative STAT3 target, *C11orf87*, showing differential hypomethylation in gastric cancer cell lines and patient samples with lower STAT3 activation status. Differential *C11orf87* methylation was also observed in gastritis, intestinal metaplasia (IM), and gastric cancer patient samples. Interestingly, our results suggested that *C11orf87* methylation can be a biomarker for early detection and disease prognosis in gastric cancer.

## Materials and methods

### Patient samples

Patient samples were collected from Changhua Christian Hospital (CCH), Taiwan or the Medical School of Zhejiang University, Hangzhou, China from March 2013 to February 2016, including 65 samples from tumor tissues and 51 patients with matched adjacent normal tissues, 27 samples from intestinal metaplasia tissues and 11 gastritis samples. The clinical-pathological data for the tissue samples are summarized in Table 1. All human studies were approved by the Institutional Review Board of the Changhua Christian Hospital, Taiwan, and the ethics committee of Zhejiang University, Hangzhou, China. The study was carried out in strict accordance with approved guidelines. Written informed consent was obtained from all participants.

### Cell culture

Gastric cancer cell line (AGS, KATO III, MKN28, MKN45, SNU1, NCI-N87, purchased from ATCC, Manassas, VA) and an immortalized gastric epithelial cell line, GES (a kind gift from Dr. Jun Yu, The Chinese University of Hong Kong, Hong Kong) were maintained in RPMI 1640 (Gibco, Waltham, MA) supplemented with 10% FBS (Gibco) and 1% Penicillin-Streptomycin (Gibco). All cells were maintained at 37˚C, with 5% $CO_2$, under a humidified incubator. Cells were treated with STAT3 inhibitor, JSI-124 (Sigma, St. Louis, MO) for 2 days and harvested for RNA extraction.

### Plasmid constructs and transfection

MKN28 cells were transiently transfected with empty vector (pcDNA3.1) or vector overexpressing a constitutively activated STAT3 mutant (STAT3c, a gift from James Darnell, Rockefeller University, NY) as previously described [9].

**Table 1. Summary of clinic-pathological data of patient samples.**

| | Tumor (n = 62) | IM (n = 27) | Gastritis (n = 8) |
|---|---|---|---|
| Age | | | |
| Median | 72.6438 | 52 | 39 |
| Range | 36.7–88.7 | 33–77 | 25–73 |
| Gender | | | |
| Female | ND | 18 | 5 |
| Male | ND | 9 | 3 |
| HP infected status | | | |
| negative | 44 | 10 | 7 |
| positive | 2 | 17 | 1 |
| ND | 16 | | |
| Histological Grade | | | |
| Low (1,2) | 23 | | |
| High (3) | 39 | | |
| Pathological Stage | | | |
| Low (I, II) | 21 | | |
| High (III, IV) | 41 | | |
| Relapse | | | |
| Primary | 40 | | |
| Recurrence | 22 | | |

ND: data not available.

## DNA extraction

DNA was extracted using a Genomic DNA Mini Kit (Geneaid, Taiwan), according to the manufacturer's instructions. DNA was then eluted in 50µl distilled water and stored at -20°C until use.

## RNA extraction and reverse-transcription PCR (RT-PCR)

Total RNA from GC cell lines or treated cell lines were extracted by Trizol (Invitrogen, Carlsbad, CA), according to the manufacturer's protocol. 1 µg of total RNA was pre-treated with DNase I (Invitrogen), and followed with cDNA synthesis by using EpiScript™ reverse transcriptase (Lucigen, Madison, WI). Reverse-transcription PCR (qRT-PCR) was performed using Platinum™ *Taq* DNA Polymerase (Invitrogen). Primers used to amplify *C11orf87* and *ACTB* cDNA can be found in Table 2.

## Infinium methylation microarray analysis

Bisulfite-modified DNA from three-paired of gastric cancer patient samples with high (n = 3) or low (n = 3) STAT3 activation, based on STAT3 IHC score, as well as AGS gastric cancer cell lines and subline depleted of STAT3 [8] were subjected to Illumina 850K methylation microarray analysis (Health GeneTech Corp, Taiwan). The methylation level of each probe (β-value) was defined by the intensity of the methylated allele (M) / (intensity of the unmethylated allele (U) + the intensity of the methylated allele (M) + 100). The microarray data has been deposited in the Gene Expression Omnibus database (accession number: GSE109541).

## Bisulfite modification and bisulfite pyrosequencing

1µg of genomic DNA was bisulfite-modified using EZ DNA Methylation Kit (ZYMO research, Orange, CA). The bisulfite modified DNA was subjected to PCR amplification using a biotin-

**Table 2. Primer sequences used in this study.**

| | Sequence 5'-3' |
|---|---|
| C11orf87 | For bisulfite pyrosequencing |
| Forward | GGTTGTTTTATTTGTTGAGTAATTTGTATT |
| Reverse | ggtcgtcagactgtcgatgaagccACCCTCTAAACACACTACCTAAACTA |
| Sequencing | AGGTTGTTGGTGGTT |
| UB04 universal primer | Ggtcgtcagactgtcgatgaagcc |
| C11orf87 | For RT-PCR |
| Forward | TCTGGTCCATTCACTCCACG |
| Reverse | TGCAGAGGCACCAGGCT |
| ACTB | For RT-PCR |
| Forward | TGCGTGACATTAAGGAGAAG |
| Reverse | GCTCGTAGCTCTTCTCCA |

labeled universal primer added as a tailed reverse primer. PCR and sequencing primers were designed using PyroMark Assay Design 2.0 software (Qiagen GmbH, Hilden, Germany). The specific primers were shown in Table 2 and were used with Invitrogen Platinum[TM] DNA Polymerase (Invitrogen) in a 25μL reaction for PCR amplification. Before pyrosequencing, 2μL of each PCR reaction was analyzed on 1.5% agarose gel. Pyrosequencing was performed on the PyroMark Q24 (Qiagen) using Pyro Gold Reagent (Qiagen) following the manufacturer's protocol. The methylation level of specific CpG sites was measured. The methylation percentage of each cytosine was determined by dividing the fluorescence intensity of cytokines with the sum of the fluorescence intensity for cytosines and thymines at each CpG site. *In vitro* methylated DNA (IVD, ZYMO research) and ultrapure distilled water (Invitrogen) was included as a positive and negative control for pyrosequencing, respectively.

## Statistical analysis

A comparison of non-parametric variables was assessed by the Mann-Whitney test. The comparison of cancer and adjacent normal was assessed by paired t-test. DNA methylation levels were performed for receiver operating characteristic curve by R package (pROC) under R 3.5.1. The cut-off for survival analyses was defined by ROC curve. Recurrence-free survival (RFS) is defined as the date of surgery to the date of recurrence or last follow-up date, while overall survival (OS) is defined as the date of surgery to the date of death or the last follow-up date. Both RFS and OS were assessed by Kaplan-Meier analysis. All statistical analysis was performed by GraphPad Prism version 5.0 for windows (GraphPad Software, La Jolla, CA, USA). $P < 0.05$ was considered as significant.

## Results

### Identification of differential methylated STAT3 target

We have previously compared the methylation profile between constitutive STAT3-activated AGS gastric cancer cells, and the subline depleted of STAT3 using Illumina 850K methylation microarray [8, 9]. Additionally, in this study, we compared the methylation profile of three pairs of gastric cancer patients, based on the STAT3 activation status. Integrative computational analyses identified several putative STAT3 targets showing promoter hypomethylation in STAT3-depleted AGS cells and gastric cancer patients showing lower STAT3 activation (as determined by STAT3 nuclear translocation). We further selected probes located within promoter CpG island, showing lower methylation in normal samples but higher methylation in

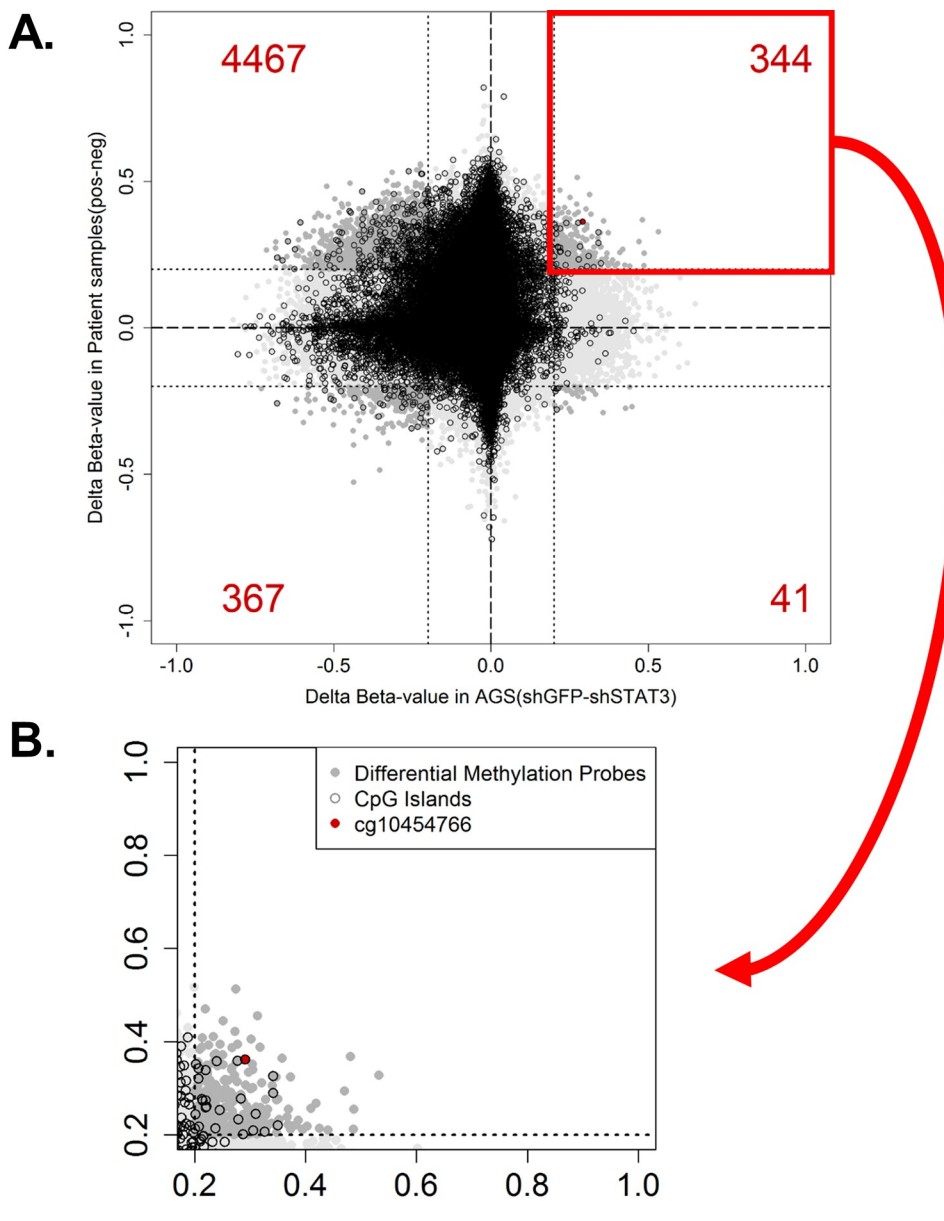

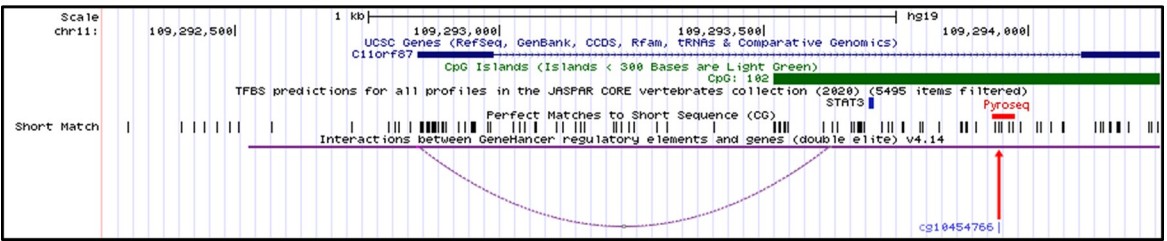

**Fig 1. Identification of cg10454766 as a STAT3-mediated hypermethylation loci.** (A) Scatter plot showing differential methylation differences in gastric cancer cell lines and patient samples. AGS gastric cancer cell lines (shGFP vs shSTAT3) and gastric cancer patients (STAT3 positive vs STAT3 negative) were subjected to Illumina 850K methylation microarray. The significant differences were defined by 20% methylation changes (0.2 delta beta-value). The amount of significant differential probes showed in red numbers for each quadrat. (B) The selected quadrat denoted STAT3-related hypermethylation. Differential methylation probes showed in dark gray. The probes

which reside within CpG island were showed in black circle. The selected probe, cg10454766, with most significant difference was shown in red dot. (C) Genomic landscape of cg10454766 (*C11orf87*) obtained from UCSC Genome browser. The predictive STAT3 binding site were performed by JASPAR2020, as shown in blue bar. The corresponding CG dinucleotides are shown in short match. The region for bisulfite pyrosequencing validation is showed in red bar, including the selected probe, cg10454766 (red arrow). Genomic interactions were performed by GeneHancer, as shown in purple curve.

tumor samples in the TCGA gastric cancer dataset. Probes with more than 20% methylation changes in both cell line and patient samples would be defined as significant differential methylation (Fig 1A). One of the probes, cg10454766, showed a significant hypomethylation in both STAT3-depleted AGS cells and patients with lower STAT3 activation (Fig 1B). Further analysis identified a putative STAT3 binding which is in close proximity to cg10454766 (242bp, Fig 1C, blue arrow), and may link to the transcription start site of *C11orf87*, as demonstrated by GeneHancer (Fig 1C, purple dash line). This probe, cg10454766, was then selected for further analysis.

## The expression and methylation of *C11orf87*

To examine the relationship between expression and methylation on *C11orf87*, we performed bisulfite pyrosequencing and RT-PCR in GC cell lines and an immortalized normal cell, GES. The results showed that those cell lines exhibited various *C11orf87* expression (Fig 2A). Unexpectedly, further bisulfite pyrosequencing also showed various *C11orf87* methylation without correlation with its expression (Fig 2B). These results suggested that *C11orf87* expression is independent of its methylation.

To investigate whether STAT3 control *C11orf87* expression, we ectopically expressed a constitutive active STAT3 mutant (STAT3c) in MKN28, a STAT3 inactive cell line [8, 9]. However, ectopic expression of STAT3c promoted *C11orf87* expression in MKN28 cells (Fig 2C), without affecting its methylation (Fig 2D). In this regard, we hypothesized that STAT3 may contribute to methylation maintenances rather than *de novo* methylation. Indeed, treatment of STAT3 inhibitor, JSI-124 [17], resulted in a downregulation of *C11orf87* expression in AGS gastric cancer cells (Fig 2E). Surprisingly, treatment of JSI-124 did not affect *C11orf87* methylation (Fig 2F). This results suggested that long term STAT3 depletion may be required to disrupt *C11orf87* methylation, as previously observed in NR4A3, a STAT3-downregulated target [9]. Taken together, these results suggested that the *C11orf87* methylation may be a passenger effect under the STAT3-mediated *C11orf87* expression.

## The *C11orf87* methylation is related to early gastric cancer initiation

As the methylation wasn't crucial for controlling *C11orf87* expression, we then investigated whether the methylation of *C11orf87* could serve as a biomarker for gastric cancer. We first analyzed the DNA methylation of cg10454766 from publicly available dataset obtained from TCGA and a previous study (GSE103186) [18], both of them were performed by Infinium HumanMethylation450 BeadChip. Overall, there was a progressive increase in DNA methylation of cg10454766 from normal gastric epithelium, mild intestinal metaplasia (IM), IM to gastric cancer (Fig 3A). Then, we performed bisulfite pyrosequencing of the CpG sites (chr11:109,293,939–109,293,966, Fig 1C, red bar) in the promoter region of *C11orf87* covering cg10454766 in our in-house samples. In agreement with TCGA and the previous study, *C11orf87* promoter showed a progressive increase in methylation upon disease progression (Fig 3B). Pairwise analysis of *C11orf87* methylation also showed higher methylation in cancer as compared to corresponding adjacent normal (Fig 3C). These results suggested that the methylation of *C11orf87* could serve as a diagnostic marker for gastric cancer.

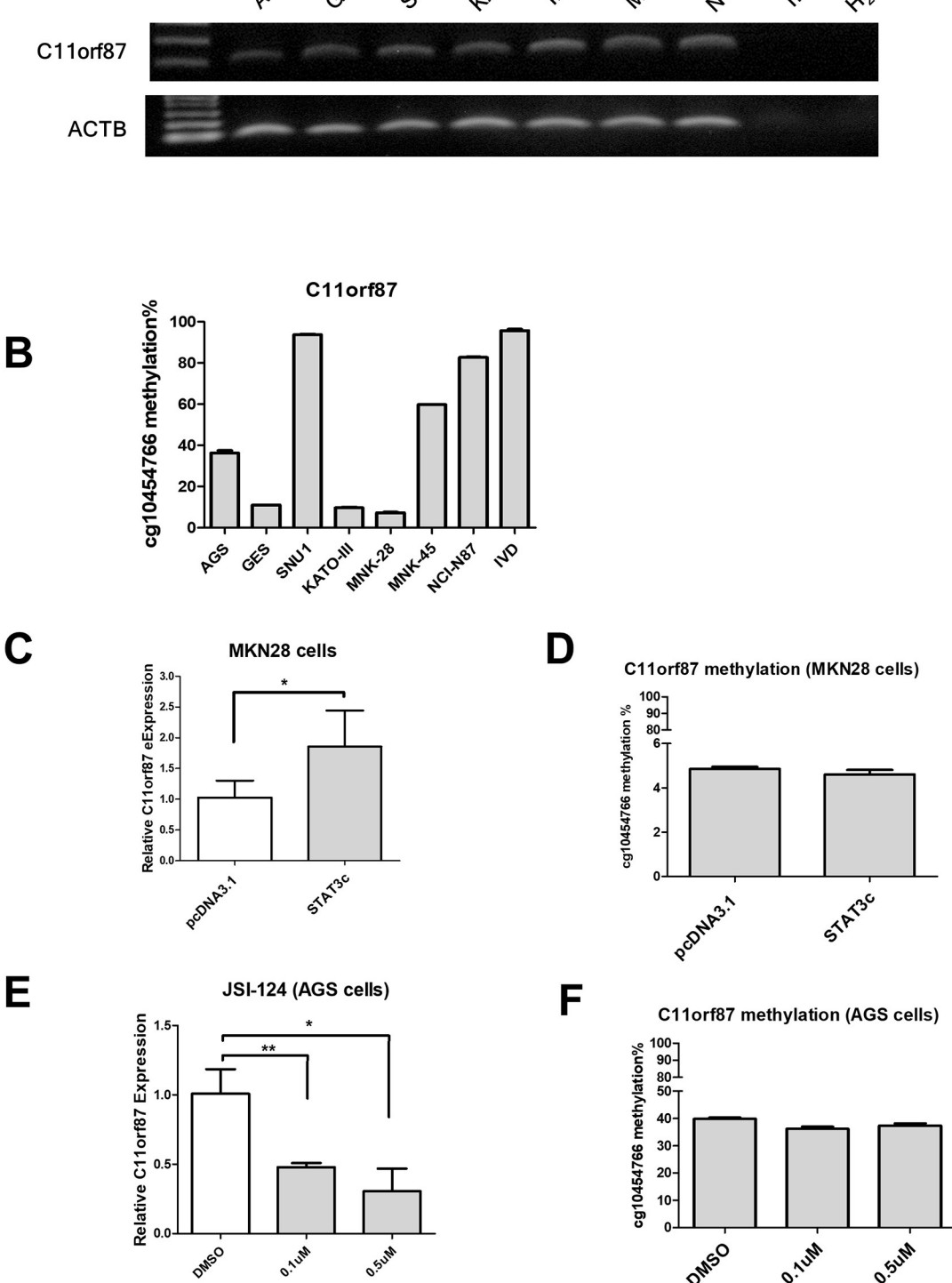

**Fig 2. The expression and methylation of *C11orf87* in gastric cancer cell lines.** (A) Expression and (B) methylation of *C11orf87* was determined by RT-PCR and bisulfite pyrosequencing, respectively in GC cell lines and an immortalized gastric epithelial cell lines, GES. MKN28 cells were transiently transfected with empty vector (pcDNA3.1) or vector expressing constitutive activated STAT3 mutant (STAT3c). *C11orf87* expression and methylation were determined by (C) qRT-PCR and (D) bisulfite pyrosequencing, respectively. AGS gastric cancer cells were treated with DMSO or STAT3 inhibitor (JSI-124) for

two days. Expression and methylation of *C11orf87* were determined by (E) qRT-PCR and (F) bisulfite pyrosequencing. Significant differences between groups were indicated by *P<0.05, **P<0.01, as determined by unpaired t-test. The original gel images can be found in S1 Fig.

For disease progression analysis, comparison of tumor grade showed that a significant higher *C11orf87* methylation was observed in patient samples with higher grade, as compared to low grade samples (Fig 4A). However, *C11orf87* methylation was not associated with tumor stage, relapse or STAT3 activity (Fig 4B–4D). As activation of STAT3 was related to poor prognosis in gastric cancer [19], these results suggested that methylation of *C11orf87* might independent to cancer progression.

## Biomarker evaluation in methylation of *C11orf87* from gastric cancer patients and non-cancerous patients

To assess if *C11orf87* methylation can be a biomarker for discriminating gastric cancer vs non-cancer, receiver operating characteristics (ROC) curve was performed using the bisulfite pyrosequencing results from our in-house samples (8 gastritis and 62 gastric cancer, Fig 5A), showing an area under curve (AUC) of 76.6%. Using a cut-off of 11.3%, the sensitivity and specificity of cancer detection is 72.6% and 87.5%, respectively.

Besides, we also combined the results from GSE103186 and TCGA gastric cancer dataset, which utilized the same microarray platform, for ROC curve analysis (Fig 5B). ROC curve using 61 normal samples from GSE103186 and 385 cancer samples from the TCGA gastric cancer dataset, showed an area under curve of 86.3%. Using a cut-off of 13.0%, the sensitivity and specificity of cancer detection as positive, the sensitivity and specificity is 70.9% and 98.4%, respectively. These results suggested that *C11orf87* methylation could be considered as a novel biomarker with well specificity for gastric cancer diagnosis.

## Hypermethylation of *C11orf87* is unexpectedly associated with better survival in gastric cancer

To further investigate the methylation of *C11orf87* for clinical outcome, we first examined the overall survival (OS) and recurrence free survival (RFS) in our in-house samples (Fig 5C and 5D). Unexpectedly, patients with higher cg10454766 methylation showed a significant better OS, but not RFS in gastric cancer. Additionally, we also performed overall survival (OS) from TCGA gastric cancer cohort (Fig 5E). In agreement with our in-house samples, patients with higher cg10454766 methylation were also associated with better OS in TCGA cohort.

## Hypermethylation of *C11orf87* is frequently observed in GI-tract cancers

We also analyzed *C11orf87* methylation in all of the gastrointestinal tract (GI) cancer from TCGA, including liver hepatocellular carcinoma (LIHC), esophageal carcinoma (ESCA), pancreatic adenocarcinoma (PAAD), stomach adenocarcinoma (STAD), Rectum Adenocarcinoma (READ) and colon adenocarcinoma (COAD). Surprisingly, *C11orf87* methylation demonstrated a hypermethylation in most of the GI-tract cancer, as compared to adjacent normal (Fig 6A). This result suggested that *C11orf87* methylation may be potentially useful for the detection of all GI cancers. In agreement with our cell lines result, there was no relationship between *C11orf87* expression and methylation in most of GI-tract cancers, except for esophageal carcinoma (ESCA) (Fig 6B). Additionally, the expression of *C11orf87* in OS from all of GI-tract cancers didn't show any correlation, which further suggested the biological role of *C11orf87* might not relate to clinical outcome (Fig 6C).

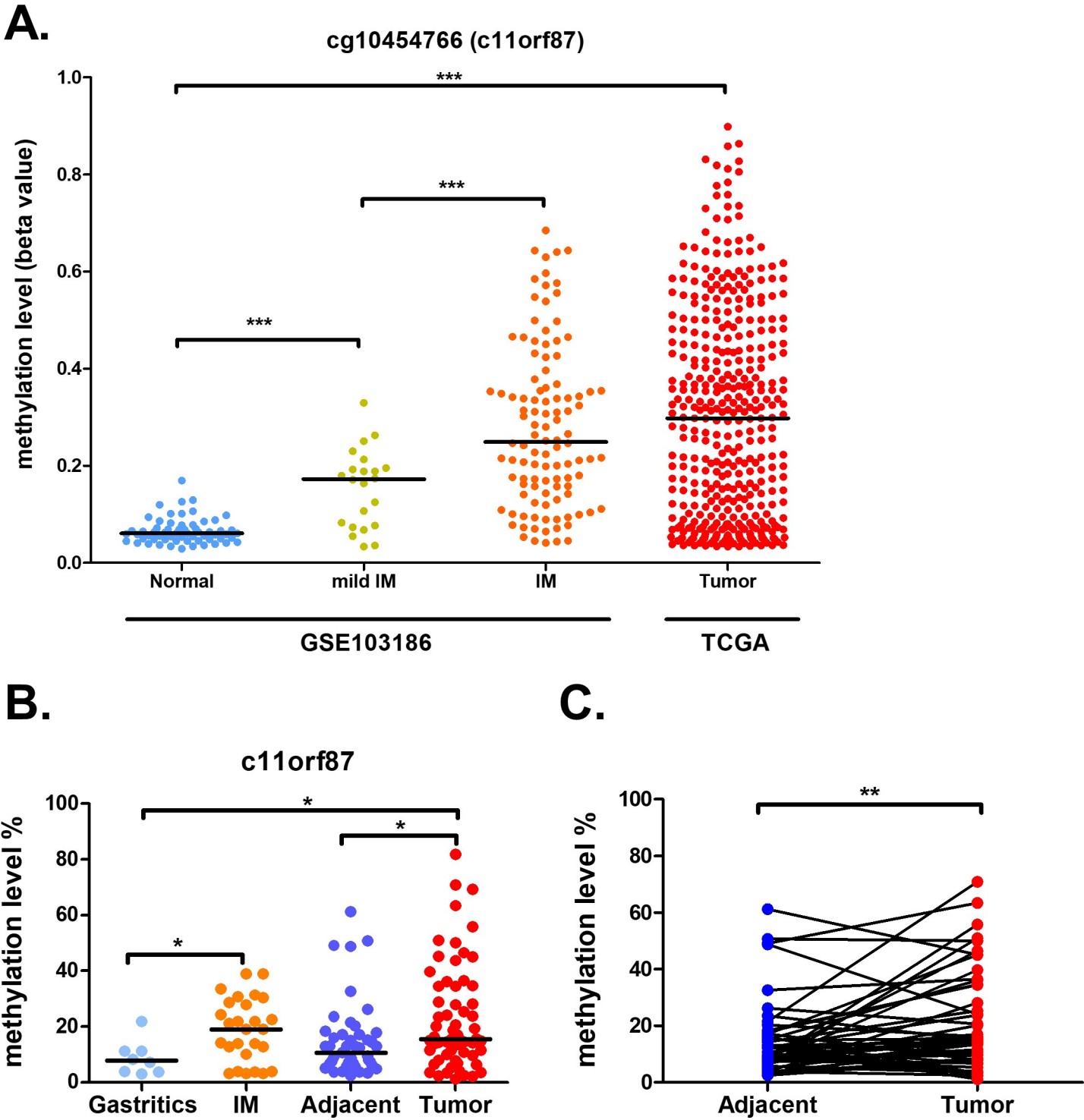

**Fig 3. Progressive hypermethylation of cg10454766 in gastric carcinogenesis.** (A) Methylation profile of cg10454766 from two datasets, GSE103186 and TCGA gastric cancer, including normal samples (n = 61), mild IM (n = 22), IM (n = 108) and cancer (n = 395). (B) Methylation of *C11orf87* from our in-house samples including gastritis (n = 8), intestinal metaplasia (n = 27), tumor-adjacent normal (n = 47) to gastric tumor patients. (n = 62), as determined by bisulfite pyrosequencing. (C) Pairwise analysis of *C11orf87* methylation in tumor-adjacent normal and tumor tissues (n = 47). Black lines denoted median values. Significant differences between groups and pairwise samples were indicated by *P<0.05, **P<0.01, ***P<0.005, as determined by Mann-Whitney U-test or paired t-test, wherever appropriate.

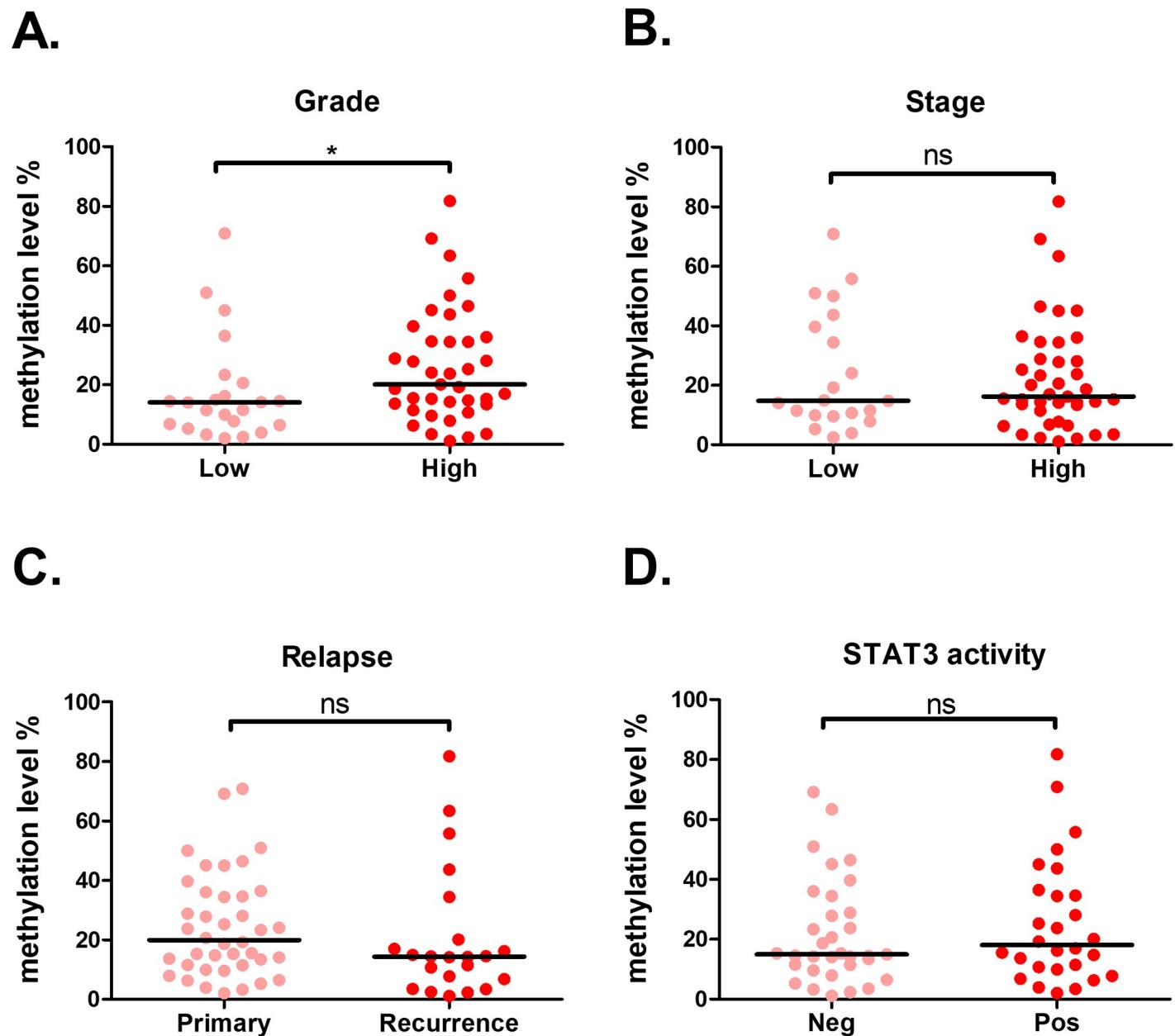

**Fig 4. Relationship between *C11orf87* methylation of cg10454766 and patients' clinical parameters.** Comparisons of *C11orf87* methylation in gastric cancer patients with different tumor (A) grade (Low: n = 23; High: n = 39), (B) stage (Low: n = 21; High: n = 41), (C) relapse (Primary: n = 40; Recurrence: n = 22), and (D) STAT3 activity (Neg: n = 32; Pos: n = 28, as determined by IHC staining). The black lines denoted median values. Significant differences between groups are indicated by *P<0.05, **P<0.01, ***P<0.005, as determined by Mann-Whitney U-test.

## Discussions

Activation of JAK/STAT signaling plays an important role in gastric carcinogenesis. In particular, STAT3 can serve as a transcriptional activator for oncogene expression [20–22]. However, the role of STAT3 as a transcriptional repressor and epigenetic regulator was less explored. By integrated experimental and bioinformatic analyses, we have previously demonstrated that depletion of STAT3 resulted in hypomethylation of STAT3 targets and tumor suppressors, *NR4A3* [9], and *SPG20* [8], in a gastric cancer cell line with constitutive activation of

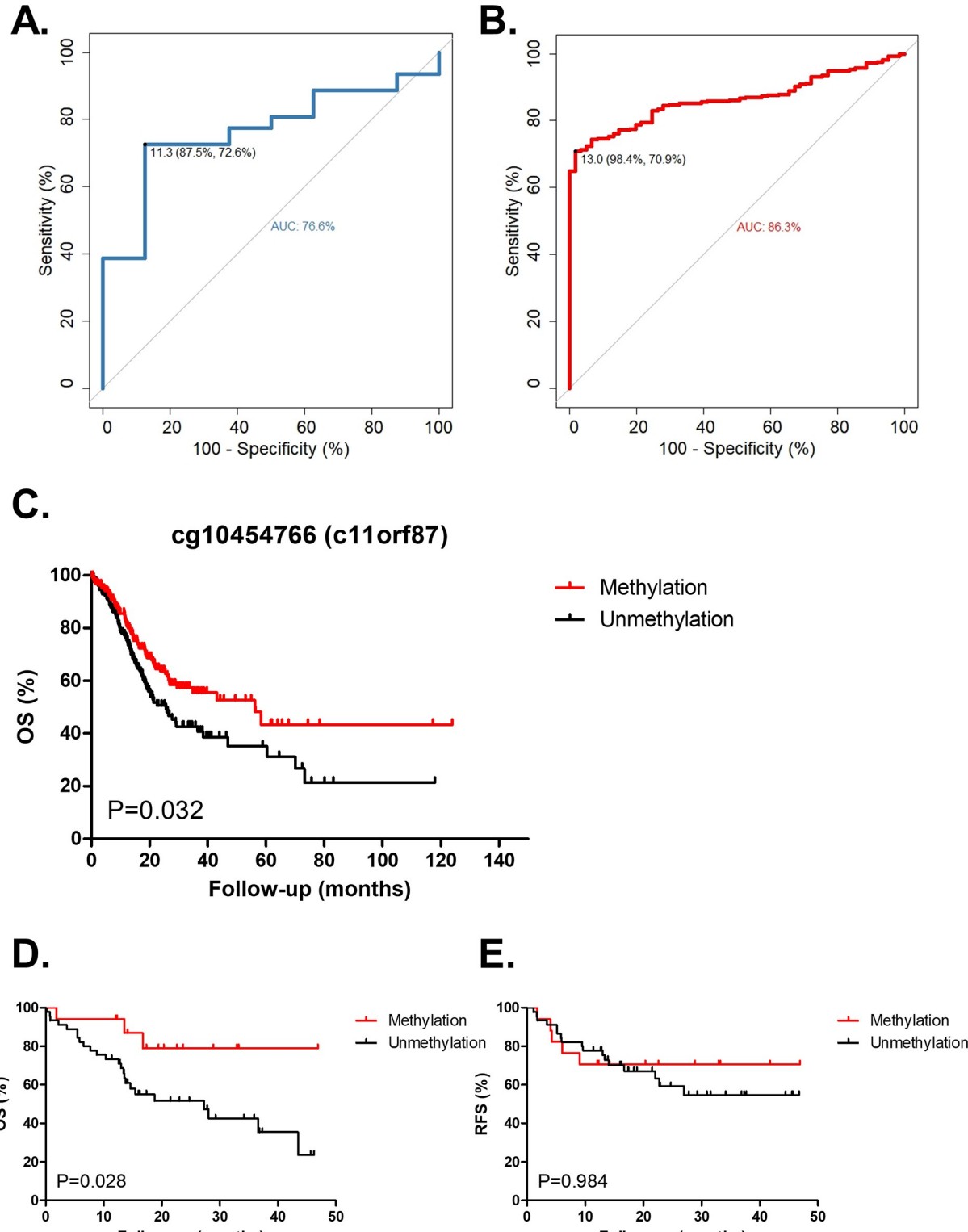

**Fig 5. The methylation of *C11orf87* served as a biomarker for diagnosis and diseases outcome.** (A) ROC curve of *C11orf87* methylation in our in-house samples, including 8 gastritis and 62 gastric cancer samples. (B) ROC curve of cg10454766 methylation in combined datasets (GSE103186 and TCGA), including 61 normal samples from GSE103186 and 385 tumor samples from TCGA. The best cut-off methylation value for specificity and sensitivity is shown (black dot). (C) Kaplan-Meier analysis of patients with differential *C11orf87* methylation status for overall survival (OS) in gastric cancer TCGA dataset (n = 385). Patients were divided into two groups according to the median value of methylation (median = 29.77%). Kaplan-Meier analysis of (D) OS and (E) recurrence-free survival (RFS) in gastric cancer patients from Taiwan. Patients were divided according to same median value from TCGA cohort. P-value is shown from the Log-rank (Mantel-Cox) test.

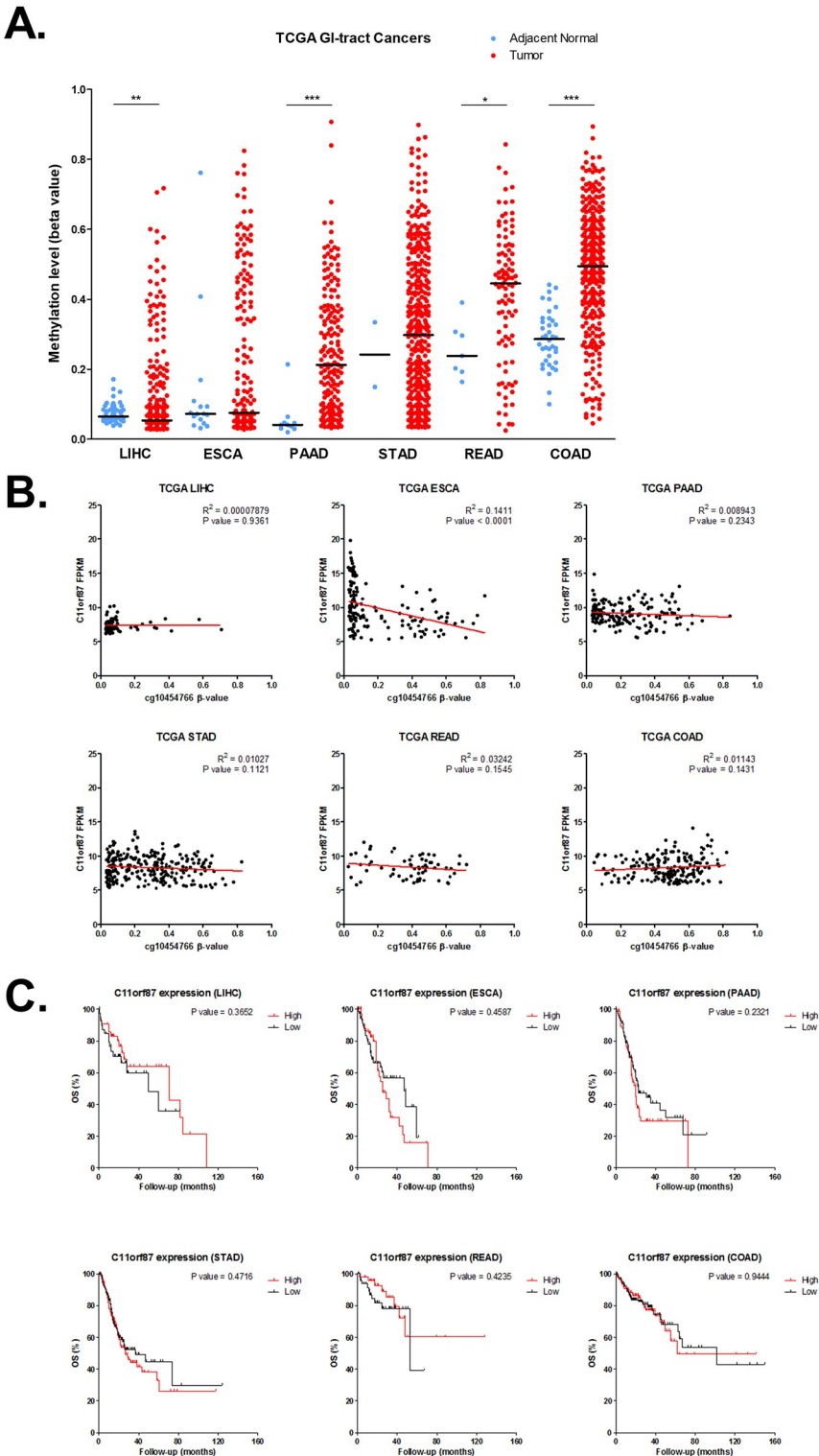

**Fig 6. The methylation and expression profiles of *C11orf87* in GI-tract cancers.** (A) Methylation of *C11orf87* in all of GI-tract cancers, including liver hepatocellular carcinoma (LIHC), esophageal carcinoma (ESCA), pancreatic adenocarcinoma (PAAD), stomach adenocarcinoma (STAD), rectum adenocarcinoma (READ), and colon adenocarcinoma (COAD). Blue dots denote adjacent normal while red dots denote cancer. Black lines denote median values. Significant differences between groups are indicated by $^*P < 0.05$, $^{**}P < 0.01$, $^{***}P < 0.005$, as determined by

Mann-Whitney U-test. (B) The relationship between *C11orf87* expression and methylation in all GI-tract cancers (LIHC: n = 84; ESCA: n = 146; PAAD: n = 160; STAD: n = 247; READ: n = 64; COAD: n = 189). (C) Kaplan-Meier analysis of patients with differential *C11orf87* expression for overall survival (OS) in GI-tract cancers from TCGA dataset (LIHC: n = 82; ESCA: n = 145; PAAD: n = 159; STAD: n = 261; READ: n = 98; COAD: n = 248). Patients were divided into two groups according to the median value of expression, while the expression value shown as 0 or missing, were excluded from the analysis.

JAK/STAT signaling. Higher *NR4A3* methylation could be observed in gastric cancer patients with STAT3 activation. Patients with higher *NR4A3* methylation were associated with poor survival. These results may be due to the recruitment of DNMT, via STAT3, to the STAT3 targets [23]. Previous studies indicated that acetylation on lysine 685 of STAT3 was crucial for the interaction with DNMT1 [24]. Targeted inhibition of STAT3 acetylation may be considered as a novel epigenetic strategy for reactivation of tumor suppressor genes in human cancer [25].

In the current study, by DNA methylation microarray, we further identified a potential STAT3 target, *C11orf87*, showing hypomethylation in AGS gastric cancer cells depleted with STAT3 and patients with lower STAT3 activation. Although STAT3 can promote the expression of *C11orf87*, there is no relationship between its expression and methylation. These results, interestingly, contradict with our previous finding that STAT3 acts as an oncogenic protein in the epigenetic silencing of tumor suppressors in gastric cancer [8, 9]. Recently, a study demonstrated that SIRT1, a histone deacetylase that participated in STAT3 deacetylation, was found to be upregulated in advanced gastric cancer [26]. The authors suggested that SIRT1 upregulation may compensate for the damaging effect induced by constitutive activation of STAT3 in gastric cancer. In this regard, SIRT1 may disrupt the interaction between STAT3 and DNMT1 by deacetylation on Lys685, which further limited methylation maintenances. Herein, we postulated that hypermethylation of *C11orf87* may serve as a "vestigial marker" for constitutive activation of STAT3 in gastric cancer. This hypothesis may further explain our clinical observation that *C11orf87* hypermethylation was related to better survival.

*C11orf87*, known as neuronal integral membrane protein 1, was found to be predominantly expressed in the brain tissue [27]. However, the involvement of *C11orf87* in human cancer has not been characterized. A recent study in head and neck cancer found that p53 mutated tumors could promote differentiation of nerve fibers, which then promoted tumor growth in this tumor microenvironment [28]. As the expression of *C11orf87* was controlled by STAT3, we, therefore, postulate that aberrant STAT3 activation may involve in promoting differentiation of nerve fibers via upregulation of *C11orf87*. Although p53 mutation is frequent in gastric cancer [29], how neuronal-related gene control gastric cancer progression still requires further investigation.

In conclusion, we are the first to demonstrate the progressive increase in *C11orf87* methylation in IM and gastric cancer. Hypermethylation of *C11orf87* was associated with better prognosis. Importantly, methylation of *C11orf87* may act as a novel diagnostic biomarker for gastric cancer. Additionally, *C11orf87* methylation is frequently observed in GI cancers, suggesting that it may also be useful for the detection of GI cancers. Besides, methylation of this potential STAT3 target may serve as a marker for the response and efficacy of targeting STAT3 in gastric cancer, as it may respect to STAT3 activity in early carcinogenesis. However, the functional and clinical role of *C11orf87* in gastric cancer warrants further investigation.

## Supporting information

**S1 Fig. The original uncropped and unadjusted gel image for Fig 2A.**
(PDF)

## Author Contributions

**Conceptualization:** Michael W. Y. Chan.

**Data curation:** Kun-Tu Yeh, Jie-Ting Low, Yin-Chen Chen, Hongchuan Jin, Shu-Hui Lin.

**Funding acquisition:** Shu-Hui Lin, Michael W. Y. Chan.

**Investigation:** Mita T. M. T. Tran, Yu-Ming Chuang, Po-Yen Hsu, Yu-Ting Lee, Yin-Chen Chen, Wan-Hong Huang.

**Resources:** Hongchuan Jin.

**Supervision:** Michael W. Y. Chan.

**Writing – original draft:** Mita T. M. T. Tran, Yu-Ming Chuang.

**Writing – review & editing:** Himani Kumari, Michael W. Y. Chan.

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
