## [Decision Letter · Decision Letter 0]

29 Jul 2020

PONE-D-20-20115

Methylomic analysis identifies C11orf87 as a novel epigenetic biomarker for GI cancers

PLOS ONE

Dear Dr. Chan,

Thank you for submitting your manuscript to PLOS ONE. After careful consideration, we feel that it has merit but does not fully meet PLOS ONE’s publication criteria as it currently stands. Therefore, we invite you to submit a revised version of the manuscript that addresses all the points raised by the two reviewers.

We look forward to receiving your revised manuscript.

Kind regards,

Qian Tao

Academic Editor

PLOS ONE

Journal Requirements:

2. In your Methods section, please provide additional information about the participant recruitment method and the demographic details of your participants. Please ensure you have provided sufficient details to replicate the analyses such as: a) the recruitment date range (month and year),  b) a description of how participants were recruited, and c) descriptions of where participants were recruited and where the research took place.

3.We note that you have indicated that data from this study are available upon request. PLOS only allows data to be available upon request if there are legal or ethical restrictions on sharing data publicly. For information on unacceptable data access restrictions, please see http://journals.plos.org/plosone/s/data-availability#loc-unacceptable-data-access-restrictions.

4.Thank you for stating the following in the Acknowledgments Section of your manuscript:

[This study was supported by grants from the Ministry of Science and Technology, Taiwan (MOST

106-2314-B-194-001-MY3; 107-2314-B-194-001; 108-2314-B-194-001; 108-2314-B-194-003-

MY2), Changhua Christian Hospital, Taiwan (106-CCH-IRP-064) and the Center for Innovative

Research on Aging Society (CIRAS) from The Featured Areas Research Center Program within

the framework of the Higher Education Sprout Project by Ministry of Education (MOE) in Taiwan.]

 [This study was supported by grants from the Ministry of Science and Technology, Taiwan (MOST 106-2314-B-194-001-MY3; 107-2314-B-194-001; 108-2314-B-194-001; 108-2314-B-194-003-MY2) to MWYC, and Changhua Christian Hospital, Taiwan (106-CCH-IRP-064) to SHL.

The funders had no role in study design, data collection and analysis, decision to publish, or preparation of the manuscript.]

Reviewers' comments:

Reviewer's Responses to Questions

**Comments to the Author**

1. Is the manuscript technically sound, and do the data support the conclusions?

Reviewer #1: Partly

Reviewer #2: Partly

2. Has the statistical analysis been performed appropriately and rigorously? 

Reviewer #1: Yes

Reviewer #2: I Don't Know

3. Have the authors made all data underlying the findings in their manuscript fully available?

Reviewer #1: Yes

Reviewer #2: Yes

4. Is the manuscript presented in an intelligible fashion and written in standard English?

Reviewer #1: Yes

Reviewer #2: No

5. Review Comments to the Author

Reviewer #1: Tran et al reported C11orf87 as a novel epigenetic biomarker for GI cancers. The authors found that C11orf87, a potential STAT3 target, was hypomethylated in gastric cancer patients and a cell line with STAT3 lower activation using DNA methylation microarray. C11orf87 was hypermethylated in GI cancers compared to their adjacent normal tissues, which was associated with better survival, thus may serve as a biomarker for GI cancers. There are a few specific issues the authors should address before publication.

Special comments:

1. The authors postulated the relationship of STAT3 activation with epigenetically silenced C11orf87 in GI cancers. However, some key experiments should be performed to identify this conclusion, such as the effect of C11orf87 methylation and expression in gastric cell lines with STAT3 reactivation.

2. The C11orf87 expression and methylation levels in GI cell lines with different STAT3 status should be examined.

3. What is the expression pattern of C11orf87 in GI tumor tissues? Is there any correlation with its methylation level?

4. As a novel biomarker of GI cancers, what are the biological functions of C11orf87 in GI cells?

Reviewer #2: 1.The detection results of different methylation sequencing methods are very different. Are the methylation detection methods used in the TCGA and GSE103186 database the same? Is the combined analysis reasonable?

2.Because the authors screened the target genes of STAT3 through DNA methylation microarray, please analyze and verify the relationship between STAT3 and C11orf87.

3.What is the expression of C11orf87 in TGCA database and in-house samples? The correlation analysis between its methylation and expression ? And, the correlation analysis between its expression and prognosis?

4.In a variety of gastrointestinal tumors, including gastric cancer, the methylation level of C11orf87 in cancer tissues is higher than that in adjacent tissues and/or normal tissues. Therefore, the author proposes that C11orf87 may be used as a biomarker for gastric cancer. However, gastric cancer patients with high C11orf87 methylation have a better prognosis, which seems to be contrary to previous results. The authors think that it may be caused by the role of STAT3 in gastric cancer, but lack of relevant experimental results.

6. PLOS authors have the option to publish the peer review history of their article (what does this mean?). If published, this will include your full peer review and any attached files.

Reviewer #1: No

Reviewer #2: No

---

## [Author Response · Author response to Decision Letter 0]

23 Nov 2020

Point-to-point responses

Comments from Reviewer #1: 

We thank the reviewer for the insightful review and interest in our manuscript. We have now carefully read each comment, and incorporated the recommendations and suggestions in the following manner:

1. The authors postulated the relationship of STAT3 activation with epigenetically silenced C11orf87 in GI cancers. However, some key experiments should be performed to identify this conclusion, such as the effect of C11orf87 methylation and expression in gastric cell lines with STAT3 reactivation.

Response: We thank the review for this important question. We now performed additional experiments to investigate the relationship between C11orf87 methylation and expression in a panel of gastric cancer cell lines. The results showed that those cell lines exhibited various C11orf87 expression (Fig. 2A). Unexpectedly, further bisulfite pyrosequencing also showed various C11orf87 methylation without correlation with its expression (Fig. 2B). These results suggested that C11orf87 expression is independent of its methylation. 

These new results can be found in Figure 2A, B and Page 7 of the result section.

2. The C11orf87 expression and methylation levels in GI cell lines with different STAT3 status should be examined.

Response: Again, we thank the reviewer for this important question. We now performed additional experiments to examine the effect of STAT3 on C11orf87 expression. We ectopically expressed a constitutive active STAT3 mutant (STAT3c) in MKN28, a STAT3 inactive cell line. However, ectopic expression of STAT3c promoted C11orf87 expression in MKN28 cells (Fig. 2C), without affecting its methylation (Fig. 2D). In this regard, we hypothesized that STAT3 may contribute to methylation maintenances rather than de novo methylation. Indeed, treatment of STAT3 inhibitor, JSI-124, resulted in a downregulation of C11orf87 expression (Fig. 2E). Taken together, these results suggested that the C11orf87 methylation may be a passenger effect under the STAT3-mediated C11orf87 expression.

These new results can be found in Figure 2C-E and Page 7 of the Result sections.

3. What is the expression pattern of C11orf87 in GI tumor tissues? Is there any correlation with its methylation level?

Response: We thank the reviewer for this question. Unfortunately, we don’t have any RNA of the in-house tumor tissue samples to perform such experiments. We therefore performed additional experiments to investigate the relationship between C11orf87 methylation and expression in a panel of gastric cancer cell lines. The results showed that those cell lines exhibited various C11orf87 expression (Fig. 2A). Unexpectedly, further bisulfite pyrosequencing also showed various C11orf87 methylation without correlation with their expression (Fig. 2B). These results suggested that C11orf87 expression is independent of its methylation. As the methylation wasn’t crucial for controlling C11orf87 expression, we then investigated whether the methylation of C11orf87 could serve as a biomarker for gastric cancer. Interestingly, we found that C11orf87 methylation can be an epigenetic biomarker for GI cancers.

4. As a novel biomarker of GI cancers, what are the biological functions of C11orf87 in GI cells?

Response: We thank the reviewer for this important question. C11orf87, known as neuronal integral membrane protein 1, was found to be predominantly expressed in the brain tissue. However, the involvement of C11orf87 in human cancer has not been characterized. A recent study in head and neck cancer found that p53 mutated tumors could promote differentiation of nerve fibers, which then promoted tumor growth in this tumor microenvironment [1]. As the expression of C11orf87 was controlled by STAT3, we, therefore, postulate that aberrant STAT3 activation may involve in promoting differentiation of nerve fibers via upregulation of C11orf87. Although p53 mutation is frequent in gastric cancer [2], how neuronal-related gene control gastric cancer progression still requires further investigation.

These statements have been added in Page 11 of the Discussion section.

Comments from Reviewer #2: 

We thank the reviewer for the insightful review and interest in our manuscript. We have now carefully read each comment, and incorporated the recommendations and suggestions in the following manner:

1.The detection results of different methylation sequencing methods are very different. Are the methylation detection methods used in the TCGA and GSE103186 database the same? Is the combined analysis reasonable?

Response: We thank the reviewer for this important question. We apologize that we didn’t state clearly the methodology of these two dataset. Methylation analysis of TCGA and GSE103186 dataset are indeed from the same microarray platform (Infinium HumanMethylation450 Beadchip). Therefore, we combined these two dataset for the analyses. We have added a statement to clarify they are indeed from the same microarray platform (Page 8 of the Result section).

2.Because the authors screened the target genes of STAT3 through DNA methylation microarray, please analyze and verify the relationship between STAT3 and C11orf87.

Response: Again, we thank the reviewer for this important question. We now performed additional experiments to examine the effect of STAT3 on C11orf87 expression. We ectopically expressed a constitutive active STAT3 mutant (STAT3c) in MKN28, a STAT3 inactive cell line. However, ectopic expression of STAT3c promoted C11orf87 expression in MKN28 cells (Fig. 2C), without affecting its methylation (Fig. 2D). In this regard, we hypothesized that STAT3 may contribute to methylation maintenances rather than de novo methylation. Indeed, treatment of STAT3 inhibitor, JSI-124, resulted in a downregulation of C11orf87 expression (Fig. 2E). Taken together, these results suggested that the C11orf87 methylation may be a passenger effect under the STAT3-mediated C11orf87 expression.

3.What is the expression of C11orf87 in TGCA database and in-house samples? The correlation analysis between its methylation and expression? And, the correlation analysis between its expression and prognosis?

Response: We thank the reviewer for this question. Unfortunately, we don’t have any RNA of our in-house tumor tissues to perform such experiments. As the methylation wasn’t crucial for controlling C11orf87 expression, we then investigated whether the methylation of C11orf87 could serve as a biomarker for gastric cancer. Interestingly, we found that C11orf87 methylation can be an epigenetic biomarker for GI cancers.

These new results can be found in Figure 2 and Page 7 of the result section.

4.In a variety of gastrointestinal tumors, including gastric cancer, the methylation level of C11orf87 in cancer tissues is higher than that in adjacent tissues and/or normal tissues. Therefore, the author proposes that C11orf87 may be used as a biomarker for gastric cancer. However, gastric cancer patients with high C11orf87 methylation have a better prognosis, which seems to be contrary to previous results. The authors think that it may be caused by the role of STAT3 in gastric cancer, but lack of relevant experimental results.

Response: We thank the reviewer for this important question. As mentioned in point #2 above, we found that ectopic expression of STAT3c promoted C11orf87 expression in MKN28 cells, without affecting its methylation. While treatment of STAT3 inhibitor, JSI-124, resulted in a downregulation of C11orf87 expression. Taken together, these results suggested that the C11orf87 methylation may be a passenger effect under the STAT3-mediated C11orf87 expression.

Recently, a study demonstrated that SIRT1, a histone deacetylase that participated in STAT3 deacetylation, was found to be upregulated in advanced gastric cancer [3]. The authors suggested that SIRT1 upregulation may compensate for the damaging effect induced by constitutive activation of STAT3 in gastric cancer. In this regard, SIRT1 may disrupt the interaction between STAT3 and DNMT1 by deacetylation on Lys685, which further limited methylation maintenances. Herein, we postulated that hypermethylation of C11orf87 may serve as a “vestigial marker” for constitutive activation of STAT3 in gastric cancer. This hypothesis may further explain our clinical observation that C11orf87 hypermethylation was related to better survival. 

We have added those statement in page 10 of the Discussion section.

References

1. Amit, M., et al., Loss of p53 drives neuron reprogramming in head and neck cancer. Nature, 2020. 578(7795): p. 449-454.

2. Rhyu, M.G., et al., Allelic deletions of MCC/APC and p53 are frequent late events in human gastric carcinogenesis. Gastroenterology, 1994. 106(6): p. 1584-8.

3. Zhang, S., et al., SIRT1 inhibits gastric cancer proliferation and metastasis via STAT3/MMP-13 signaling. J Cell Physiol, 2019. 234(9): p. 15395-15406.

---

## [Decision Letter · Decision Letter 1]

23 Dec 2020

PONE-D-20-20115R1

Methylomic analysis identifies C11orf87 as a novel epigenetic biomarker for GI cancers

PLOS ONE

Dear Dr. Chan,

Thank you for submitting your manuscript to PLOS ONE. After careful consideration, we feel that it has merit but does not fully meet PLOS ONE’s publication criteria as it currently stands. Therefore, we invite you to submit a revised version of the manuscript that addresses all the points raised by the two reviewers.

We look forward to receiving your revised manuscript.

Kind regards,

Qian Tao

Academic Editor

PLOS ONE

Reviewers' comments:

Reviewer's Responses to Questions

**Comments to the Author**

1. If the authors have adequately addressed your comments raised in a previous round of review and you feel that this manuscript is now acceptable for publication, you may indicate that here to bypass the “Comments to the Author” section, enter your conflict of interest statement in the “Confidential to Editor” section, and submit your "Accept" recommendation.

Reviewer #1: (No Response)

Reviewer #2: (No Response)

2. Is the manuscript technically sound, and do the data support the conclusions?

Reviewer #1: Partly

Reviewer #2: (No Response)

3. Has the statistical analysis been performed appropriately and rigorously? 

Reviewer #1: Yes

Reviewer #2: (No Response)

4. Have the authors made all data underlying the findings in their manuscript fully available?

Reviewer #1: Yes

Reviewer #2: (No Response)

5. Is the manuscript presented in an intelligible fashion and written in standard English?

Reviewer #1: Yes

Reviewer #2: (No Response)

6. Review Comments to the Author

Reviewer #1: In the revised manuscript, the authors demonstrated that C11orf87 methylation could be an epigenetic biomarker for GI cancers, and its expression is independent of its methylation. There are only a few comments for the authors which they should take into account before publication.

Comments:

1. The labeling is misplaced in Figure 2A.

2. The authors identified a potential STAT3 target, C11orf87, through using DNA methylation microarray, which is hypomethylated in gastric cancer patients with lower STAT3 activation and AGS cells with STAT3 inactivation. The authors thus inferred a possible correlation between C11orf87 methylation/expression and STAT3 activation. However, after ectopic expression of STAT3c, no methylation changes of C11orf87 was found in MKN28 with STAT3 inactivation (Figure 2D), and also no bars were shown. The authors should repeat the experiment and do statistical analysis (Figure 2B, 2D). The methylation analysis of C11orf87 in cells with the treatment of STAT3 inhibitor JSI-124 should also be examined.

Reviewer #2: 1.Fig2a. there are 9 samples,but only 8 bands for C11orf87. 7 binds for Actin.

2.Fig2c,2d. please detect expression of C11orf87 by qRT-PCR.

7. PLOS authors have the option to publish the peer review history of their article (what does this mean?). If published, this will include your full peer review and any attached files.

Reviewer #1: No

Reviewer #2: No

---

## [Author Response · Author response to Decision Letter 1]

26 Feb 2021

Point-to-point responses

Comments from Reviewer #1: 

We thank the reviewer for the insightful review and interest in our manuscript. We have now carefully read each comment, and incorporated the recommendations and suggestions in the following manner:

1. The labeling is misplaced in Figure 2A.

Response: We apologize for this mistake, we have now aligned the labeling in the correct place. 

2. The authors identified a potential STAT3 target, C11orf87, through using DNA methylation microarray, which is hypomethylated in gastric cancer patients with lower STAT3 activation and AGS cells with STAT3 inactivation. The authors thus inferred a possible correlation between C11orf87 methylation/expression and STAT3 activation. However, after ectopic expression of STAT3c, no methylation changes of C11orf87 was found in MKN28 with STAT3 inactivation (Figure 2D), and also no bars were shown. The authors should repeat the experiment and do statistical analysis (Figure 2B, 2D). The methylation analysis of C11orf87 in cells with the treatment of STAT3 inhibitor JSI-124 should also be examined.

Response: We thank the reviewer for this important question. We have repeated the bisulfite pyrosequencing in Fig 2B and D, and the results are very consistent, showing low SD. We have replaced Fig 2B and D with error bars. Regarding AGS cells treated with STAT3 inhibitor (JSI-124), we have now performed bisulfite pyrosequencing to examine the changes of C11orf87 methylation. Surprisingly, treatment of JSI-124 did not affect C11orf87 methylation in AGS cells (Fig. 2F). This results suggested that long term STAT3 depletion may be required to disrupt C11orf87 methylation, as previously observed in NR4A3, a STAT3-downregulated target [1].

This new result can be found in Fig 2F and Page 7 of the Result section. 

References

1. Yeh CM, Chang LY, Lin SH, Chou JL, Hsieh HY, et al. (2016) Epigenetic silencing of the NR4A3 tumor suppressor, by aberrant JAK/STAT signaling, predicts prognosis in gastric cancer. Sci Rep 6: 31690.

Comments from Reviewer #2: 

We thank the reviewer for the insightful review and interest in our manuscript. We have now carefully read each comment, and incorporated the recommendations and suggestions in the following manner:

1. Fig2a. there are 9 samples, but only 8 bands for C11orf87. 7 binds for Actin.

Response: We apologize for this mistake, we have now aligned the labeling in the correct place. 

2.Fig2c,2d. please detect expression of C11orf87 by qRT-PCR.

Response: Expression of C11orf87 in Fig 2C and E is now replaced by qRT-PCR. Thank you for this comment.

---

## [Editor Report · Decision Letter 2]

8 Apr 2021

Methylomic analysis identifies C11orf87 as a novel epigenetic biomarker for GI cancers

PONE-D-20-20115R2

Dear Dr. Chan,

We’re pleased to inform you that your manuscript has been judged scientifically suitable for publication and will be formally accepted for publication once it meets all outstanding technical requirements.

Kind regards,

Qian Tao

Academic Editor

PLOS ONE
---

## [Editor Report · Acceptance letter]

13 Apr 2021

PONE-D-20-20115R2 

Methylomic analysis identifies *C11orf87* as a novel epigenetic biomarker for GI cancers 

Dear Dr. Chan:

I'm pleased to inform you that your manuscript has been deemed suitable for publication in PLOS ONE. Congratulations! Your manuscript is now with our production department. 

Kind regards, 

on behalf of

Dr. Qian Tao 

Academic Editor

PLOS ONE